# The Importance of Diet in Predicting the Remission of Urticaria—Determination of Allergen-Specific IgE

**DOI:** 10.3390/medicina57070679

**Published:** 2021-07-01

**Authors:** Monica Iuliana Ungureanu, Liliana Sachelarie, Radu Ciorap, Bogdan Aurelian Stana, Irina Croitoru, Tudor Ovidiu Popa

**Affiliations:** 1Department of Preventive Medicine and Interdisciplinarity, Faculty of Medicine, Grigore T. Popa University of Medicine and Pharmacy-Iasi, 700115 Iași, Romania; monica.ungureanu@umfiasi.ro (M.I.U.); irina.croitoru@umfiasi.ro (I.C.); 2Department of Biophysics, Faculty of Dental Medicine, Apollonia University, 700511 Iași, Romania; 3Department of Biomedical Sciences, Faculty of Medical Bioengineering, Grigore T. Popa University of Medicine and Pharmacy-Iasi, 700115 Iași, Romania; 4Department of Morpho-Functional Sciences II, Faculty of Medicine, Grigore T. Popa University of Medicine and Pharmacy-Iasi, 700115 Iași, Romania; aurelian.stana@umfiasi.ro; 5“Sf. Maria” Clinical Pediatric Hospital Iaşi, 700309 Iași, Romania; 6Department of Surgery II, Faculty of Medicine, Grigore T. Popa University of Medicine and Pharmacy-Iasi, 700115 Iași, Romania; tudor.popa@umfiasi.ro

**Keywords:** food allergens, urticaria, allergen-specific IgE, children

## Abstract

*Background and Objectives:* Different types of food introduced gradually in the diet will expose children to different food allergens, increasing the chance of developing allergic diseases. The aim of our study was to determine if allergen-specific IgE values can influence, depending on the diet, the prediction of remission of urticaria in children. *Materials and Methods:* This prospective study was conducted in 132 patients diagnosed over two years with urticaria, admitted to “Sf. Maria” Clinical Pediatric Hospital Iaşi. Total IgE assay was performed by ELISA, and determination of specific serum IgE by the CLA System Quanti Scan method (Innogenetics, Heiden, Germany). Data were gathered and statistical analysis was performed using statistical software SPSS, using descriptive and inferential statistics. *Results:* The determination of specific IgE to food allergens was performed on a total of 132 cases. The values of specific IgE were positive for one or more food allergens in 84 patients (63.64%). The most common allergens involved were: cow’s milk in 33.3% cases, egg white in 22.6% cases, and hazelnuts in 11.9% cases. The specific IgE values for the different types of food included in our study had a predictive value for disease remission. *Conclusions:* The determination of specific IgE confirms the presence of a particular food allergen and may have predictive value for the future development of an allergic manifestation.

## 1. Introduction

Lifestyle changes (diet, excessive food processing), exposure to irritants and pollutants, lead to changes in the body’s immune reactivity, resulting in an increased prevalence of allergic diseases.

Based on current information on molecular and cellular biology, histopathology, symptom score, and impact on quality of life, the current trend is to consider urticaria as a systemic disease and not as an organ disease. This approach to the condition will have an impact on the recommended therapeutic and prophylactic behavior. Awareness precedes or accompanies the early symptoms of allergy, and its identification can be considered a marker for the introduction of prophylactic measures. The identification of the food factors involved as well as the therapeutic possibilities present important research objectives. Studies on urticaria triggered by food allergens place its onset at an early age, when there are difficulties in establishing the diagnosis: atypical clinical forms, variations in age-dependent sensitization markers, reduced compliance for skin tests, or challenge tests. Early identification of allergic sensitization in children with familial atopic terrain is essential in avoiding exposure to allergens and introducing specific drug treatment or immunotherapy. An immune mechanism-specific antigen triggers food allergy after exposure to a food allergen [1,2,3]. In children, food allergy occurs more frequently in cow’s milk, egg white, cereals, and fish. Its incidence is higher in the first three years of life. Later, after the age of four, the respiratory allergy is diagnosed more frequently [4]. In this paper, we made a correlation between dietary factors and IgE specific allergen values.

## 2. Materials and Methods

We conducted a prospective study over a period of two years on a group of 132 patients admitted to the Second Pediatric Clinic Emergency Clinical Hospital for Children Iași. The study aimed to establish correlations between diet, immune response, and urticaria in children.

Criteria for inclusion in the study were clinical diagnosis of urticaria, suggestive anamnesis of patients who had one or more episodes of acute urticaria in which a causal link with a dietary factor was suspected, or atopic family background.

The study protocol was approved by the Ethics Commission of the University of Medicine and Pharmacy “Grigore T. Popa” Iasi. The patient’s informed consent was made known to the patient and the child’s parents or legal guardians. In carrying out the study, national legislation and European directives on the protection of human rights and human dignity in the application of biology and medicine were complied with (Law 17/22.02.2001, Law 206/2004, Oviedo Convention). The evaluation of patients was performed by obtaining epidemiological, anamnestic, general clinical examination, laboratory examinations (hematological, biochemical, immunological), and skin prick tests. The study protocol followed the clinical, biological, and immunological aspects longitudinally as well as the specific recommended therapeutic conduct. 

Patients were evaluated at the first hospitalization, then periodically at variable intervals depending on the evolution, when the presence/absence of urticaria and the specific IgE level were assessed. The determination of specific IgE was performed at the beginning as a baseline and after six months. Only foods in which the results showed specifically positive IgE were excluded from the diet. The biological investigations performed consisted of routine analyses and immunological investigations. Total IgE assay was performed by ELISA, and determination of specific serum IgE for 15 of the most common food allergens and aeroallergens (nine food allergens and six aeroallergens) by the CLA System Quanti Scan method (Innogenetics, Heiden, Germany).

Specific aeroallergen IgE determination was performed for Graminaceous mixture pollen gx6: Cynodondactylon, Loliumperenae, Sorghum halepense, Bromusinermis, Holcuslanata, Paspalum notatum; Dermatophagoides pteronyssinus; Dermatophagoides farina; Ambrosia elatior pollen; dog hair, and cat hair.

A total of 53 skin prick tests were performed, of which 52.63% were positive.

Statistical analysis was performed using statistical software SPSS v.20 (SPSS Inc., Chicago, IL, USA), and data were analyzed using descriptive and inferential statistics. Statistically significant differences were considered for a 95% confidence interval, using parametric or nonparametric tests. Student’s *t*-test, Pearson-χ^2^, Fisher’s test, ANOVA test, linear regression, logistic regression, and Cox model regression analysis were used. The significance threshold was *p* = 0.05 (defined as sufficient in the literature). For continuous variables, descriptive statistical indicators were calculated (mean, standard deviation, standard error, minimum, maximum, and quartile intervals). Total IgE values do not show a homogeneous distribution, so the Kruskal–Wallis test specific to this type of data was applied to compare the values according to the age of the patients. Based on IgE values and the value of the disease activity score, ROC (receiver operating characteristic) curves were performed. Based on AUC values (area under the curve), it was possible to evaluate the predictability of IgE specific to each food allergen studied.

## 3. Results

The determination of specific IgE to food allergens was performed on a total of 132 cases.

Age distribution is detailed in Figure 1, where mean age was 5.83 years, with most frequent cases in age group of one year with 15.2% of cases and in age group 0–1 year with 14.4% of cases.

Gender distribution was almost equal: 70 cases representing 53% were boys, and 62 cases representing 47% were girls.

The values of specific IgE were positive for one or more food allergens in 84 cases (63.64%). The determination of specific IgE in food allergens was performed in 132 cases. Specific IgE values were positive for one or more food allergens in 84 cases (63.64%). The most common allergens involved were: cow’s milk (33.3% cases), egg white (22.6% cases), hazelnuts (11.9% cases), fish (9.5% cases), soy (7.1% cases), nuts (7.1% cases), wheat flour (6% cases), strawberries 92.4% cases), and rye flour (1.2% case). Statistical indicators of specific IgE vs. allergen type are shown in Table 1 and the specific IgE values according to the type of allergen in Figure 2.

### 3.1. Evaluation of Specific IgEas Predictor of Disease Remission

Based on specific IgE values, ROC (receiver operating characteristic) curves were performed for each type of food. For specific IgE values to be considered predictable for disease remission, the area under the ROC curve (AUC) must have values greater than 0.6. According to these values, the significance level of the test, for a confidence interval considered to be 95%, must be lower than the critical reference threshold of 0.05.

Specific IgE values of wheat flour have a predictive value on disease remission. The results showed significant AUC values (AUC_IgEflour-wheat specific_ = 0.674, *p* = 0.038, 95% CI: AUC → 0.507–0.840) (Table 2, Figure 3).

Table 3 and Table 4 compare the specific IgE values at baseline with the values measured six months later, after the exclusion of the allergen from the diet (dynamic values).

It is observed that specific IgE-Egg White Values Decreased Significantly (t = 9.29, *p* << 0.01, 95% CI).

Specific IgE egg white values had a moderate value for prediction on disease remission as the results showed AUC values (AUC_eggwhiteIgE_ = 0.461, *p* = 0.617, 95% CI: AUC → 0.289–0.633) (Table 5, Figure 4).

Specific IgE values of cow’s milk had a predictive value on disease remission, and the results showed values close to 0.6, considered predictable [4] (AUC_IgE cow’s milk_ = 0.528, *p* = 0.715, 95% CI: AUC → 0.366–0.691) (Table 6, Figure 5).

### 3.2. Specific IgE Values in Dynamics

The mean values in dynamics for specific IgE of hazelnuts, strawberries, and fish are given in Table 6.

Specific IgE values of strawberries, (t = 981, *p* = 0.022, 95% CI), hazelnuts (t = 8.99, *p* << 0.01, 95% CI), and fish (t = 6.17, *p* = 0.00045, 95% CI) decreased significantly (Figure 6).

The mean values in dynamics for specific IgE of rye flour, soy, and nuts are described in Table 7.

Specific IgE values in the dynamics of cow’s milk (t = 7.95, *p* << 0.01, 95% CI), rye flour (t = 5.64, *p* = 0.254, 95% CI), soy (t = 3.51, *p* = 0.017, 95% CI), and nuts (t = 5.2, *p* = 0.0034, 95% CI) decreased significantly (Figure 7).

Skin prick tests were performed in 53 patients and of these, 52.63% were positive.

Correlation of skin prick tests with the IgE levels showed a good correlation between results. Thus, significantly higher values were observed in the case of positive tests (H_Kruskal-Wallis_ = 6.88, *p* = 0.038, 95% CI) (Table 8, Figure 8).

Polysensitivity to food allergens and aeroallergens was correlated with respiratory allergic manifestations (asthma and allergic rhinitis), occurring predominantly in the age group over four years, while atopic dermatitis was correlated with food allergy in children under four years.

## 4. Discussion

The specific IgE level varies depending on age, total serum IgE level, amount and duration of allergen exposure. In addition, the specific IgE level can be altered by exposure to cross-reactive allergens and by specific immunotherapy. There are studies that have correlated the level of specific IgE in food allergens with the risk of an allergic reaction in case of re-exposure to the incriminated allergen. The determination of specific IgE confirms the sensitization to a certain allergen and may have a predictive value for the future development of an allergic manifestation. The presence of specific food allergens in the first year of life is associated with an increased risk of pneumalergen sensitization, with the subsequent onset of a respiratory allergic condition. The analysis of IgE levels in different age groups (0–2 years, 2–6 years, and over 6 years) showed that total IgE levels are lower in older children compared to those in young patients.

Sichereret et al. described in their study that food allergies, defined as an immune response to food proteins, affect about 8% of children and 2% of adults [5], but the prevalence is increasing, as is the case with all allergic diseases. In the case of food allergies, the mechanisms involved may be IgE-mediated or non-IgE-mediated immunological reactions. The correlation of skin prick tests with speIgE values established a correlation of the obtained results. Skin tests have a sensitivity of 95% and a specificity of 50%.

Predictive factors regarding the resolution of allergy are: the characteristics of the initial reaction (isolated urticaria/angioedema versus other manifestations), the specific IgE level, the intensity of the skin reaction to the skin prick test, and the severity of atopic dermatitis.

A negative skin test result has a high predictive value in children older than one year, while in children less than one year, its negative predictive value is lower due to the particularities of cutaneous mast cells (small number and low reactivity).

The clinical tolerance criterion was the absence of symptoms for six months from the time of reintroduction of the food. In the present study, oral tolerance was installed in 38 children, more frequently for cow’s milk and egg white.

Dietary treatment in children with food allergies are aimed at eliminating allergenic food. The evaluation of the parameters that can determine the recurrence of urticaria by multivariate analysis showed that specific IgE values, compliance with diet, and treatment can be considered predictive factors for the recurrence of urticaria.

Knowing the risk factors for recurrence allows patients to be included in the risk group and in patients with atopic dermatitis, the immune status must be evaluated in order to highlight an immune deficiency.

There are a number of determinants involved in the prognosis of food allergies: ethnicity, sex, type of food, innate immune system, challenge dose, sensitization status, composition of the intestinal microbiome, and the presence of comorbidities [5,6].

Establishing the prognostic factors and the phenotype of food allergies is important in the prevention and management of these diseases. A more rigorous classification of allergic patients will make it possible to determine the severity of the exclusion diet and the timing of the reintroduction of the forbidden food [7]. Regarding egg allergy, some studies estimate that it affects about 0.5–2.5% of children [8]. This food ranks second in frequency in food allergies. The resolution rate of egg allergy is slow, as most studies have shown that egg tolerance develops at three years, but there are also studies that showed that about half of the children developed tolerance at the age of 12 years. There is a persistence of egg allergy in 42% of adolescent children, suggesting that the number of adults with egg allergy may increase over time, although current estimates of adult egg allergy were 0.2% [9,10]. The determination of specific IgE in egg white (ovalbumin and ovomucoid) and the basophil activation test showed a decrease in specific IgE values and specific antigen activation after the induction of food tolerance [11].

Induction of food tolerance is a specific process for each food, which is mediated by immunological changes, decreased specific IgE values, and specific spontaneous activation of basophils.

Ruinemas-Koerts et al. and Sicherer et al. conducted studies on egg allergy by measuring the allergen specific component, ovalbumin specific IgE, using two methods: indirect luminescent oxygen channeling immunoassay (LOCI) and enzyme-linked immunosorbent assay (ELISA) [9]. The comparison of the results obtained by the two methods was performed by ROC (receiver operator characteristic) and the conclusion was that the LOCI method was superior in accuracy to the ELISA method. The basophil activation test can also be used to diagnose and predict allergy, for example, in a study on cow’s milk allergy, which was performed both at the initial diagnosis and to determine tolerance to this food. This type of test has high sensitivity and specificity (86–100%).

Sicherer and Leung conducted a study on the diagnosis of food allergies by performing the oral food challenge (OFC) test and showed that 29% of the tests performed were positive, with most food allergens being peanuts, eggs, and milk [12,13,14]. Serum levels of IgE specific for hazelnuts and eggs were significantly different in cases with allergic OFC compared to non-allergic OFC, and the clinical manifestations of urticaria, angioedema, and vomiting were correlated with the allergic OFC test. Food allergens more frequently involved in the occurrence of allergic manifestations differ depending on the geographical area and eating habits. A recent study in Spain using the measurement of specific IgE and skin tests indicated a prevalence of 30.92% of allergic manifestations associated with food intake, of which 2.92% were caused by garlic and onions [8,15]. A study on the prevalence of food allergies in Iraq in patients aged 13 to 52 years showed that the most commonly involved food allergens were seafood (13.15%), fish (12.76%), a mixture of cereals (grain mix 9.21%), chicken (7.44%), garlic and sesame (5.31%) [16]. In our study, the most commonly involved food allergens were cow’s milk, egg white, and hazelnuts.

Regarding hazelnut allergy, recent research considers it to be useful in determining the diagnosis to determine the epitopes for the response to allergens in hazelnuts, along with the basophil activation test and the oral loading test [17,18].

Limitations and Strengths

This study has several limitations. The determination of specific IgE for food allergens and the dynamic tracking of the specific IgE level can be an important landmark regarding the subsequent evolution towards an IgE mediated hypersensitivity reaction. Specific IgE determinations were performed for only 15 more frequently involved allergens and also did not perform the basophil activation test that could have been useful in the diagnosis and prediction of allergy.

Through this study, we want to educate and train patients on food allergies, as well as to increase the responsibility of parents, starting with the newborn stage, when natural nutrition must be strongly promoted, and continuing with the subsequent stages in which, for the children with food allergies, the recommendations of a regime of eviction the incriminated allergens must be followed.

## 5. Conclusions

Dynamic monitoring of specific IgE levels can provide information on the evolution of allergic manifestations. The decrease in the specific IgE level compared to the allergenic proteins precedes the installation of the immunological tolerance, and a sustained high level of specific IgEis associated with the persistence of the allergic phenomena. In the studied group, the dynamic tracking of specific IgE to food allergens showed a progressive decrease after the implementation of a removal program of the suspected allergen.

Increasing parental responsibility for children’s nutrition must be emphasized and encouraged, especially for those with a familial atopic background, due to the fact that the prevalence of food allergies is increasing, which has a significant impact on pediatric patients.

The diagnosis of food allergy is largely reliant on medical history and tests for sensitization, but allergen specific IgE specific use can improve diagnostic accuracy and predict the response to the different therapeutic measures.

## Figures and Tables

**Figure 1 medicina-57-00679-f001:**
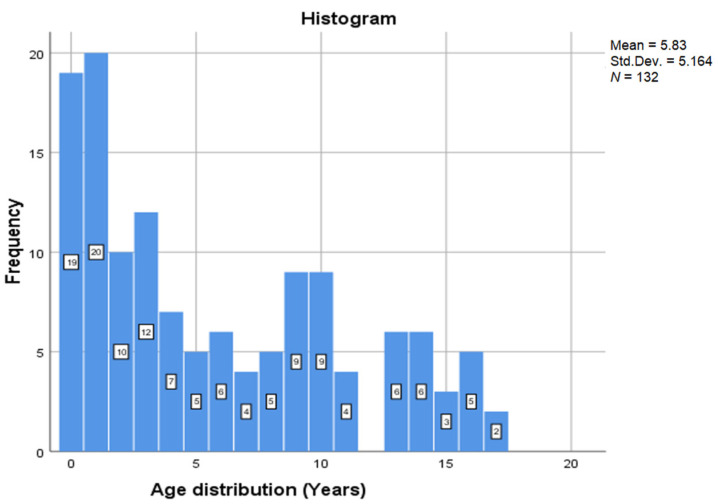
Age distribution.

**Figure 2 medicina-57-00679-f002:**
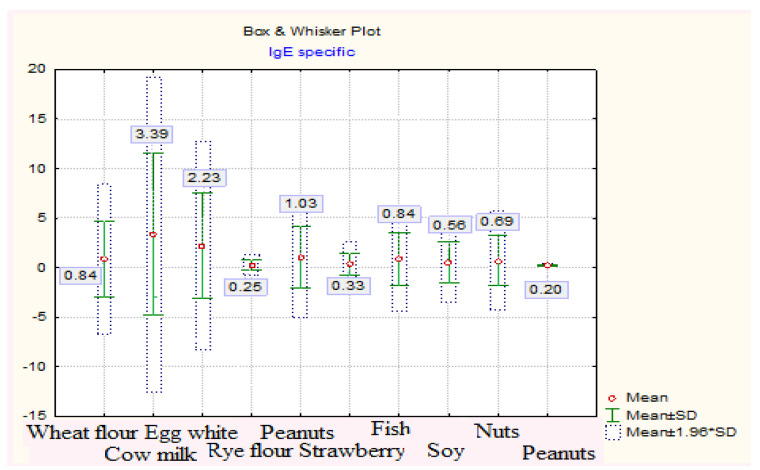
Specific IgE values depending on the type of allergen.

**Figure 3 medicina-57-00679-f003:**
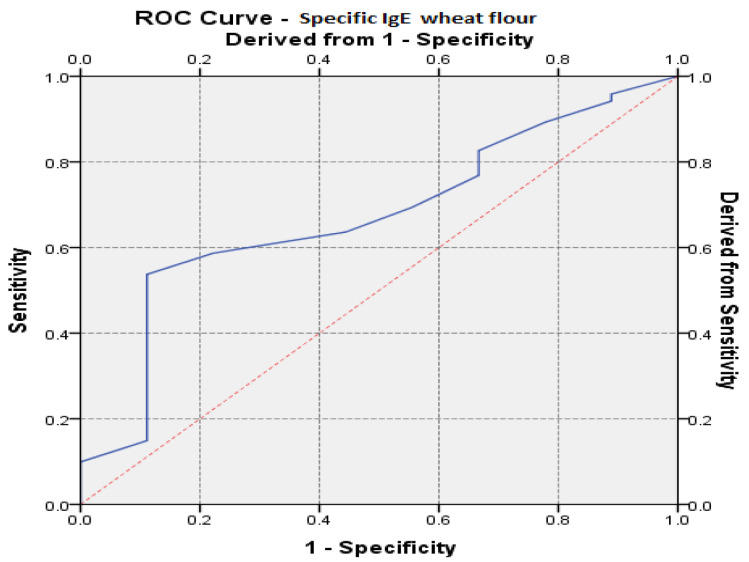
The ROC (receiver operating characteristic) curve regarding the IgE prediction specific to wheat flour vs. remission of the disease.

**Figure 4 medicina-57-00679-f004:**
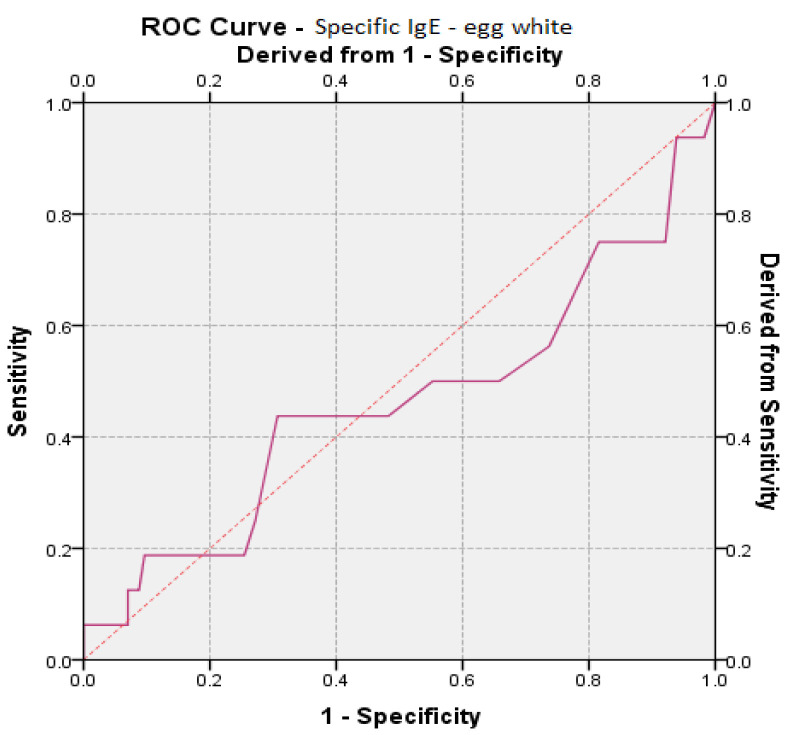
ROC (receiver operating characteristic) curve for egg white specific IgE prediction vs. remission of the disease.

**Figure 5 medicina-57-00679-f005:**
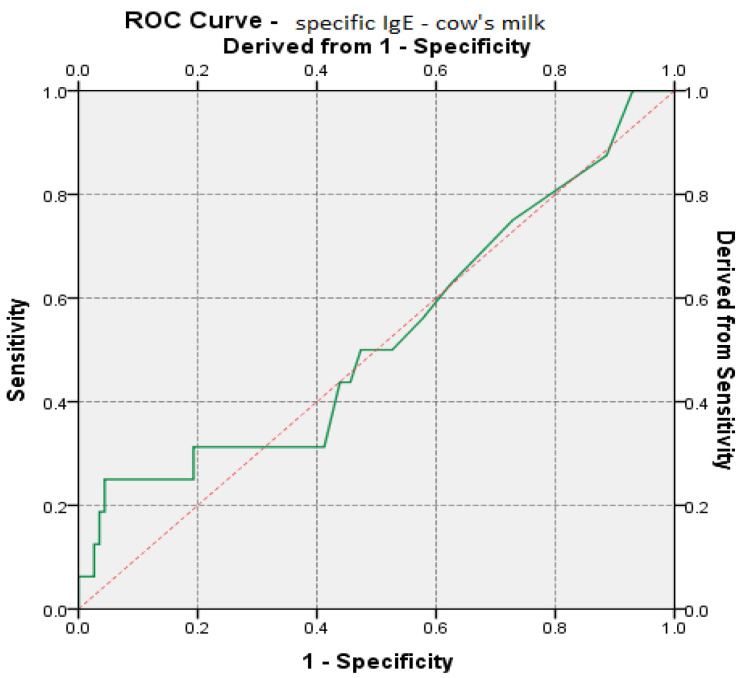
ROC (receiver operating characteristic) curve regarding IgE prediction specific to cow’s milk vs. remission of the disease.

**Figure 6 medicina-57-00679-f006:**
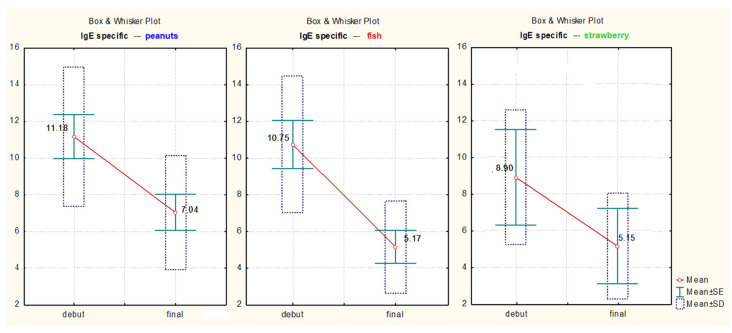
Specific IgE values in dynamics for strawberry, peanuts, and fish.

**Figure 7 medicina-57-00679-f007:**
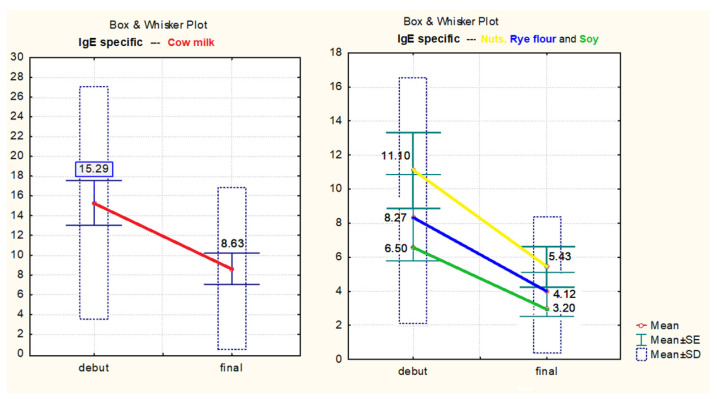
Specific IgE values in dynamics of cow milk, nuts, rye flour, and soy.

**Figure 8 medicina-57-00679-f008:**
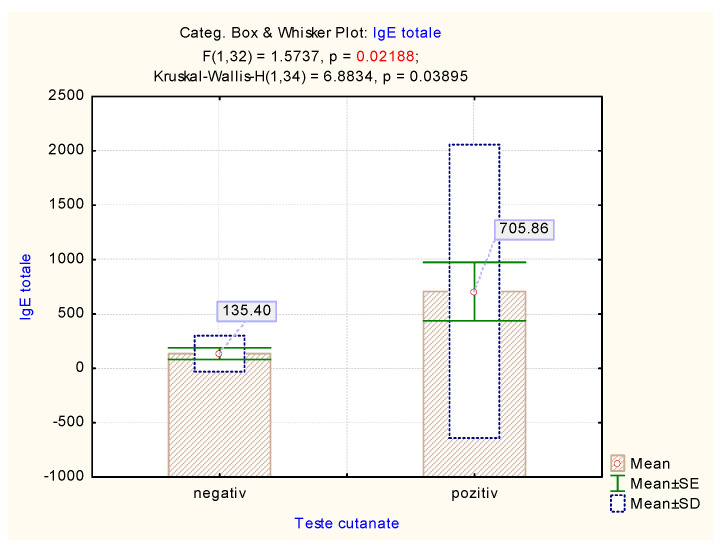
The mean value of total IgE vs. skin prick test results.

**Table 1 medicina-57-00679-t001:** Statistical indicators of specific IgE vs. allergen type.

Title Allergen	Average IgE Specific	Media		Dev.std	Er.std	Min	Max	Q25	Median	Q75
		−95%	+95%							
wheat flour	0.84	0.18	1.50	3.83	0.33	0.10	36.30	0.15	0.19	0.27
Cow’s milk	3.39	1.99	4.79	8.11	0.71	0.10	51.30	0.14	0.20	0.31
egg white	2.23	1.31	3.16	5.36	0.47	0.10	21.60	0.16	0.20	0.30
rye flour	0.25	0.16	0.35	0.55	0.05	0.10	6.50	0.15	0.19	0.27
hazelnuts	1.03	0.50	1.56	3.08	0.27	0.10	16.80	0.14	0.18	0.27
strawberries	0.33	0.14	0.52	1.12	0.10	0.10	11.50	0.14	0.18	0.27
fish	0.84	0.38	1.30	2.67	0.23	0.10	16.20	0.15	0.18	0.28
soy	0.56	0.21	0.92	2.07	0.18	0.10	18.90	0.15	0.19	0.25
nuts	0.69	0.26	1.13	2.52	0.22	0.10	21.50	0.14	0.19	0.27
peanuts	0.20	0.19	0.22	0.07	0.01	0.10	0.35	0.14	0.19	0.27

**Table 2 medicina-57-00679-t002:** Estimated parameters in the analysis of the ROC curve for specific IgE-wheat flour/disease remission.

Area under the ROC Curve					
Variables Tested Factors	Area under the Curve (AUC)	Standard Error	Level of Significance. ^b^ (*p*)	AUC 95% Reference Interval	
				Inf. Limit	Upper Limit
Specific IgE-wheat flour	0.674	0.085	0.0383	0.507	0.840

^b^. Null hypothesis: area = 0.5.

**Table 3 medicina-57-00679-t003:** Comparison of dynamic mean values for specific IgE of wheat flour and egg white.

*t*-Test for Dependent Samples								
	Mean	Std.Dv.	N	Diff.	Std.Dv.-Diff.	t	df	*p*
Wheat flour	15.29259	11.73761						
Wheat flour 2	8.63333	8.15867	27	6.659259	4.347524	7.959133	26	0.000000
Egg white	14.92222	4.874128						
Egg white 2	9.71667	4.505585	18	5.205556	2.377213	9.290416	17	0.000000

**Table 4 medicina-57-00679-t004:** Comparison of dynamic mean values for specific IgE of hazelnuts, strawberries, and fish.

*t*-Test for Dependent Samples								
	Mean	Std.Dv.	N	Diff.	Std.Dv.-Diff.	t	df	*p*
hazelnuts	11.18000	3.808120						
hazelnuts 2	7.04000	3.115267	10	4.140000	1.455411	8.995278	9	0.000009
strawberries	8.900000	3.676955						
strawberries 2	5.150000	2.899138	2	3.750000	0.777817	9.818182	1	0.022710
fish	10.75000	3.722135						
fish 2	5.17500	2.518361	8	5.575000	2.554408	6.173047	7	0.000457

**Table 5 medicina-57-00679-t005:** Estimated parameters in the analysis of the ROC curve for specific IgE-egg white/disease remission.

Area under the ROC Curve					
Variables Tested Factors	Area under the Curve (AUC)	Standard Error	Level of Significance. ^b^ (*p*)	AUC 95% Reference Interval	
				Inf. Limit	Upper Limit
Specific IgE-egg white	0.461	0.088	0.617	0.289	0.633

^b^. Null hypothesis: area = 0.5.

**Table 6 medicina-57-00679-t006:** Estimated parameters in the analysis of the ROC curve for specific IgE-cow’s milk/disease remission.

Area under the ROC Curve					
Variables Tested Factors	Area under the Curve (AUC)	Standard Error	Level of Significance. ^b^ (*p*)	AUC 95% Reference Interval	
				Inf. Limit	Upper Limit
specific IgE-cow’s milk	0.528	0.083	0.715	0.366	0.691

^b^. Null hypothesis: area = 0.5.

**Table 7 medicina-57-00679-t007:** Comparison of average values in dynamics for specific IgE of rye flour, soy, and nuts.

*t*-Test for Dependent Samples								
	Mean	Std.Dv.	N	Diff.	Std.Dv.-Diff.	t	df	*p*
rye flour	6.5	0.02						
rye flour 2	3.2	0.07		3.300000	2.377213	5.6415	1	0.0254
soy	8.266667	6.137643						
soy 2	4.116667	3.619622	6	4.150000	2.893959	3.512621	5	0.017053
nuts	11.10000	5.445365						
nuts 2	5.43333	2.919361	6	5.666667	2.665833	5.206793	5	0.003448

**Table 8 medicina-57-00679-t008:** The test for a comparison of total IgE values vs. skin prick test results.

**Kruskal–Wallis Test**	**H (95% Confidence Interval)**	***p***
	6.8834	0.03895

## Data Availability

The data are available from the authors upon reasonable request.

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
