# Peer review of "The Importance of Diet in Predicting the Remission of Urticaria—Determination of Allergen-Specific IgE"

_medicina, 2021, doi:10.3390/medicina57070679_

Round 1

Reviewer 1 Report

In the studied group, the dynamic tracking of specific IgE to food allergens showed a progressive decrease after and the implementation of a removal program of the suspected allergen.

Increasing parental responsibility for children’s nutrition must be emphasized and encouraged, especially for those with familial atopic background, due to the fact that the prevalence of food allergies is increasing.

The diagnosis of food allergy is largely reliant on medical history and tests for sensitization, but allergen specific IgE specific use can improve diagnostic accuracy and predict the response to the different therapeutic measures.

Dynamic monitoring of specific IgE levels can provide information on the evolution of allergic manifestations. 

The decrease of the specific IgE level compared to the allergenic proteins precedes the installation of the immunological tolerance, and a sustained

Q1 : Title

Is clear, short length, descriptive and well done

Q2 : Affiliation data, Abstract and Keywords

 The affiliation of the authors must be completed, indicating the place or working and the country when they live

An structured abstract is designed and performed and contains an useful and complete information about the importance of several foods can play in relation to the remission of urticaria in children

In the Keywords, must be added “children”, because is the studied population

Q3. 1. Introduction

 It is an important point to consider urticaria as a systemic disease and not as an organ disease and the early identification of allergic sensitization in children is very relevant

Food allergy in children shows a higher incidence in the first 3 years of life

  1. Materials and Methods

A prospective study over a period of 2 years, on a group of 132 pts. Was done and contains a large population to be examined

The study protocol was complete and approved by the ECUM of the hospital

Patients were evaluated at the first hospitalization, then periodically at variable intervals depending of the presence/absence of urticaria and the specific IgE level were assessed by ELISA

The determination of specific IgE was performed at the beginning as a baseline and after 6 months. Only foods in which the results were positive for IgE, were excluded from the diet.

The biological investigations were determined on specific serum IgE for 9 food allergens and 6 aeroallergens

53 skin prick test were performed, of which 52,63% were positive

Statistical analysis was performed and the specific tests for qualitative and quantitative data must be described and also the p level of comparison will be described

  1. Results

The age distribution is a little right-sided because the percentange of children less than 3 years old is relatively small

Specific IgE values were positive for one or more allergens in 84 cases (63,64%)

I don´t understand the meaning of “over” included in Table 1. Please explain it.

Specific IgE values- wheat flour have predictive value on disease remission

Specific IgE-egg white values, decreased significantly on disease remission (t=9.29, p<<0.01, 95%CI).

Specific IgE values - strawberries, (t = 981, p = 0.022, 95% CI), hazelnuts (t = 8.99, p << 0.01, 95% CI) and fish (t = 6.17, p = 0.00045, 95% CI) decreased significantly (Fig.6). Correlation of skin prick tests with the IgE levels showed a good correlation between results. Thus, significantly higher values are observed in the case of positive tests (HKruskal-Wallis=6.88, p=0.038, 95%CI). (Table 8, Fig. 8)

Polysensitivity to food allergens and aeroallergens was correlated with respiratory allergic manifestations (asthma and allergic rhinitis), occurring predominantly in the age  group over 4 years, while atopic dermatitis was correlated with food allergy in children under 4 years

  1. Discussion

The specific IgE level varies depending on age, total serum IgE level, amount and duration of allergen exposure. Also, the specific IgE level can be altered by exposure to cross-reactive allergens and by specific immunotherapy.

The analysis of IgE levels in different age  groups (0-2 years, 2-6 years and over 6 years) showed that total IgE levels are lower in older children, compared to those in young patients.

Sichereret all described that food allergies, defined as an immune response to food proteins, affect about 8% of children and 2% of adults but the prevalence is increasing, as is the case with all allergic diseases.

A negative skin test result has a high predictive value in children older than 1 year, while in children less than 1 year its negative predictive value is lower, due to the particularities of cutaneous mast cells (small number and low reactivity).

Regarding egg allergy, some studies estimate that it affects about 0.5-2.5% of children. This food ranks second in frequency in food allergies.

The resolution rate of egg allergy is slow, most studies show that egg tolerance develops at 3 years, but there are also studies that show that about half of children develop tolerance at the age of 12 years.

The determination of specific IgE in egg white (ovalbumin and ovomucoid) and the basophil activation test, show a decrease in specific IgE values and specific antigen activation after induction of food tolerance

Food allergens more frequently involved in the occurrence of allergic manifestations differ depending on the geographical area and eating habits. 

  1. Conclusions

In the studied group, the dynamic tracking of specific IgE to food allergens showed a progressive decrease after and the implementation of a removal program of the suspected allergen.

Increasing parental responsibility for children’s nutrition must be emphasized and encouraged, especially for those with familial atopic background, due to the fact that the prevalence of food allergies is increasing.

The diagnosis of food allergy is largely reliant on medical history and tests for sensitization, but allergen specific IgE specific use can improve diagnostic accuracy and predict the response to the different therapeutic measures.

Dynamic monitoring of specific IgE levels can provide information on the evolution of allergic manifestations. 

The decrease of the specific IgE level compared to the allergenic proteins precedes the installation of the immunological tolerance, and a sustained high level of specific IgEis associated with the persistence of the allergic phenomena

  1. References

They are well selected, good in quality and enough in total number (19)

Author Response

Dear Madam/Sir,

First of all, we would like to thank the Reviewer for his efforts in a thorough and insightful review of our article entitled "The importance of diet in predicting the remission of urticaria in children - determination of allergen-specific IgE. "

Your critical remarks allowed the authors to refine almost all the work details, making it more readable and better organized.

The corrections we have made will be related directly to the form in which they were delivered:

  1. The affiliation of the authors must be completed, indicating the place of work and the country when they live

RESPONSE1: We completed the affiliation of the authors

  1. In the Keywords, must be added "children," because is the studied population

RESPONSE2: We add" children" in the Keywords

  1. Statistical analysis was performed and the specific tests for qualitative and quantitative data must be described and also the p level of comparison will be described

RESPONSE3: In the" Materials and Methods" paragraph, we add the following text:

Statistically significant differences were considered for a 95% confidence interval, using parametric or nonparametric tests. Student's t-test, Pearson-χ2, Fisher's test, ANOVA test, linear regression, logistic regression, and Cox model regression analysis were used. The significance threshold was p = 0.05 (defined as sufficient in the literature). For continuous variables, descriptive statistical indicators were calculated (mean, standard deviation, standard error, minimum, maximum, and quartile intervals). Total IgE values do not show a homogeneous distribution, so the Kruskal-Wallis test specific to this type of data was applied to compare the values according to the age of the patients. Based on IgE values and the value of the disease activity score, ROC (Receiver Operating Characteristic) curves were performed. Based on AUC values (Area Under the Curve), it was possible to evaluate the predictability of IgE specific to each food allergen studied.

  1. I don't understand the meaning of "over" included in Table 1. Please explain it.

RESPONSE4: It was a mistake. We corrected this. The right word is"fish", like is showed in figure 2.

The second Reviewer made the following observation: ”The authors did not only study urticaria in children but allergy in general, so the title should be changed.”

We decided to change the title to:“The importance of diet in predicting the remission of urticaria - determination of allergen-specific IgE. “

 I remain most respectfully yours,

Prof.dr. Liliana Sachelarie

Reviewer 2 Report

Children with food allergies are at risk of allergic reactions that range from mild to potentially life-threatening. The condition can contribute to reduced quality of life and barriers to participation in day-to-day activities. Although firm prevalence data are lacking, there is a strong impression that food allergy has increased, and rates as high as approximately 10% have been documented. Genetic, epigenetic, and environmental risk factors are being elucidated increasingly, creating the potential for improved prevention and treatment strategies targeted to those at risk.  
It is very important to conduct research and publish the results of the prevalence of allergies in children. The tests were performed with methods generally recognized as correct. The results were properly processed by selecting appropriate statistical methods. Only the title of the article does not reflect the research carried out. The authors did not only study urticaria in children but allergy in general, so the title should be changed.

Author Response

(The authors gave the same response as above.)
